# Cap-proximal nucleotides via differential eIF4E binding and alternative promoter usage mediate translational response to energy stress

Ana Tamarkin-Ben-Harush[1], Jean-Jacques Vasseur[2], Françoise Debart[2], Igor Ulitsky[3†], Rivka Dikstein[1*†]

[1]Department of Biomolecular Sciences, The Weizmann Institute of Science, Rehovot, Israel; [2]Department of Nucleic Acids, IBMM UMR 5247, CNRS-Université Montpellier-ENSCM, Montpellier, France; [3]Department of Biological Regulation, The Weizmann Institute of Science, Rehovot, Israel

**Abstract** Transcription start-site (TSS) selection and alternative promoter (AP) usage contribute to gene expression complexity but little is known about their impact on translation. Here we performed TSS mapping of the translatome following energy stress. Assessing the contribution of cap-proximal TSS nucleotides, we found dramatic effect on translation only upon stress. As eIF4E levels were reduced, we determined its binding to capped-RNAs with different initiating nucleotides and found the lowest affinity to 5'cytidine in correlation with the translational stress-response. In addition, the number of differentially translated APs was elevated following stress. These include novel glucose starvation-induced downstream transcripts for the translation regulators eIF4A and Pabp, which are also translationally-induced despite general translational inhibition. The resultant eIF4A protein is N-terminally truncated and acts as eIF4A inhibitor. The induced Pabp isoform has shorter 5'UTR removing an auto-inhibitory element. Our findings uncovered several levels of coordination of transcription and translation responses to energy stress.

*For correspondence: rivka.dikstein@weizmann.ac.il

†These authors contributed equally to this work

Competing interests: The authors declare that no competing interests exist.

## Introduction

Transcription start site (TSS) selection and alternative promoter (AP) usage increase transcriptome diversity and its regulation. For example the level of transcription initiation can vary between different transcription start sites (TSS) under different growth conditions, in response to a specific signal or in different cell types and tissues. In addition, mRNA isoforms with different 5' leaders can vary in their translation efficiency or their half-lives. Likewise, AP usage can lead to the generation of protein isoforms that differ in their N-termini and as a result have different or even opposite biological functions. Recent large-scale promoter analysis in hundreds of human and mouse primary cell types shed light on the prevalence of AP usage in mammals (*Forrest et al., 2014*). Several studies have examined translation and stability of transcript isoforms of the same gene (*Arribere and Gilbert, 2013*; *Floor and Doudna, 2016*; *Wang et al., 2016*) but little is known about the contribution of AP usage to the translational response to stress.

The process of protein synthesis is highly energy consuming and tightly regulated by the availability of nutrients, oxygen and growth factors. Downregulation of the translation machinery is a major mechanism that allows cells to preserve energy and cope with environmental deficiencies. Under these conditions translation of many mRNAs is inhibited but that of others is unchanged or even

**eLife digest** The production of new proteins is a complex process that occurs in two steps known as transcription and translation. During transcription, the cell copies a section of DNA to make molecules of messenger ribonucleic acid (or mRNA for short) in the nucleus of the cell. The mRNA then leaves the nucleus and enters another cell compartment called the cytoplasm, where it serves as a template to make proteins during translation. A mRNA molecule contains a sequence of building blocks known as nucleotides. There are four different types of nucleotides in mRNA and the order they appear in the sequence determines how the protein is built.

Both transcription and translation consume a lot of energy so they are highly regulated and sensitive to environmental changes. However, since transcription and translation happen in different cell compartments, it is not known if and how they are coordinated under stress.

Tamarkin-Ben-Harush et al. studied transcription and translation in mouse cells that were starved of glucose. The experiments show that the identity of the very first nucleotide in the mRNA – which is dictated during transcription – has a dramatic influence on the translation of the mRNA, especially when the cells are starved of glucose. This first nucleotide affects the ability of a protein called eIF4E, which recruits the machinery needed for translation, to bind to the mRNA. The experiments also show that there is a dramatic increase in the number of distinct mRNAs that are transcribed from the same section of DNA but translated in a different way during glucose starvation.

The findings of Tamarkin-Ben-Harush et al. show that transcription and translation are highly coordinated when cells are starved of glucose, allowing the cells to cope with the stress. The next step is to further analyze the data to find out more about how transcription and translation are linked.

enhanced in order to survive the stress. The translation inhibition response is mediated by several mechanisms, in particular by the impairment of key initiation factors eIF2 and eIF4E (*Sonenberg and Hinnebusch, 2009*). Stresses such as growth factor, energy and amino acid deficiencies affect the formation of the cap binding complex eIF4F, comprising of the initiation factors eIF4E, eIF4G and eIF4A. Under these conditions, eIF4E-Binding Proteins (4EBPs) which bind eIF4E with high affinity and interfere with its binding to eIF4G, is activated by dephosphorylation resulting in inhibition of cap-dependent translation. 4EBP is controlled by the mammalian target of rapamycin (mTOR), a protein kinase that phosphorylates and diminishes its ability to bind eIF4E. Under circumstances of limited nutrients or other stresses mTOR activity is inhibited, 4EBP activity is enhanced and cap-dependent translation is suppressed (*Sonenberg and Hinnebusch, 2009*). The effect of the suppression of the general translation initiation factors under stress appears to vary from gene to gene, and is dictated by specific regulatory elements present in the mRNAs. A well-characterized mRNA feature associated with strong translational inhibition is the TOP element (5' Terminal Oligo Pyrimidine), an uninterrupted stretch of 4–15 pyrimidines, starting with cytidine at the most 5'end of the mRNA (for a review see *Meyuhas and Kahan [2015]*). A large fraction of TOP mRNAs code for proteins that are associated with translation and they are strongly translationally repressed following various physiological stresses that inhibit mTOR signaling by a mechanism that is not fully understood. It is thought that specific factors that bind the 5' polypyrimidine track mediate the positive and negative translation regulation of the TOP mRNAs. While several features of cellular mRNAs involved in specific translational response to metabolic stress were characterized, the impact of specific TSS usage on the cellular response to stress is poorly investigated.

In the present study we aimed to obtain a global view of the effects of TSS selection on translation following metabolic energy stress. To this end, we combined polysomal profiling with quantitative assessment of the 5' ends of mRNAs. By comparing transcript isoforms that differ in their 5' end we identified hundreds of genes with APs that are differentially translated, in particular following energy stress, suggesting that a major determinant of the differential response is associated with transcription-induced APs. We also determined the contribution of TSS nucleotides to translation. Strikingly, while the cap-proximal nucleotides have no significant effect on translation under optimal growth conditions, they display dramatic effects on the translational response to stress. We

demonstrate that eIF4E levels drop following the stress and that the binding affinity of eIF4E towards capped mRNA with different first nucleotide varies significantly, with a relatively lower affinity to 5' polypyrimidine as in TOP mRNAs. We next characterized two genes encoding translation regulatory factors with differentially translated APs. The first is eIF4A, the helicase subunit of the cap complex eIF4F, in which we identified a novel glucose-starvation-induced intronic promoter. The induced isoform, which has distinct 5'UTR and initiating AUG, is efficiently translated in energy deficient cells in spite of the global translation inhibition. The resultant protein is N-terminally truncated and acts as an eIF4A inhibitor, most likely to facilitate the stress response. The second is poly-A binding protein (Pabp, Pabpc1) in which repression of the major TOP-containing isoform following stress is coupled with the induction of a downstream TSS that generates an mRNA isoform with a much shorter 5'UTR that is highly translated. Interestingly, the induced isoform lacks the well-characterized Pabp auto-inhibitory element (*Sachs et al., 1986*; *Bag and Wu, 1996*; *Wu and Bag, 1998*; *Hornstein et al., 1999*). Our findings expand the understanding of the regulatory mechanisms that coordinate the cellular response to metabolic energy stress both in transcription and translation.

## Results

### Marked increase in differential translation of alternative promoters following stress

To obtain a global and quantitate view of the impact of transcription start site (TSS) selection on translation efficiency following cellular stress we combined polysomal profiling with quantitative assessment of the 5' ends of mRNAs (*Figure 1A*) (note that ribosomal footprinting is not suitable for our purpose since it primarily records coding sequences). Mouse Embryonic Fibroblasts (MEFs) were subjected to glucose deprivation, a stress that causes inhibition of global translation at the initiation and elongation levels mediated by signaling pathways that are AMPK-dependent (*Bolster et al., 2002*; *Dubbelhuis and Meijer, 2002*; *Krause et al., 2002*; *Reiter et al., 2005*; *Shenton et al., 2006*) and independent (*Inoki et al., 2003*; *Kalender et al., 2010*; *Sinvani et al., 2015*). Cells were then lysed and subjected to sucrose gradient sedimentation. As expected, the ribosome profile in response to glucose starvation (GS) was dramatically changed as the relative amount of 80S monoribosome was increased while polysomal fractions were decreased (*Figure 1B*), indicating global inhibition of translation. The fractions from the gradient were merged to create three major pools: Polysome-free, from the top of the gradient to a single ribosome; Light, two to five ribosomes per mRNA; and Heavy, six or more ribosomes per mRNA. RNA was extracted from the pooled fractions and equivalent RNA volume was taken from each fraction for transcription start site library preparation. At this stage, two RNA spikes, GFP and Luciferase (transcribed and capped in vitro) were added into each fraction pool, to serve as controls for sample handling. For TSSs sequencing, we used the CapSeq method as previously described (*Gu et al., 2012*). In this method an adaptor is added only to the cap site of the mRNA and is used for deep sequencing. Thus the sequence information reports on the 5' end usage of transcripts within the polysomal profile. Samples from two independent biological replicates were subjected to Illumina sequencing. The 5' ends of mapped reads predominantly corresponded to annotated and CAGE-defined transcription start sites, showing that our methodology provides quantitative single-base resolution view of the TSS landscape (*Figure 1C*).

When the results of two independent experiments were combined and normalized using internal controls, 9286 promoters that corresponded to 6861 genes had at least 500 reads per promoter and were considered in further analysis (*Supplementary file 1*). Under these thresholds, three quarters of the genes had a single expressed promoter in these cells and the rest had two or more (*Figure 1D*). To validate that the TSS reads are in accordance with the expected biological outcome we first analyzed the global translational and transcriptional response to GS. We summed all the reads from each fraction for all the promoters and found that the overall translation was reduced upon GS as evident from the increase and decrease of the polysome-free (Free) and polysomal fractions (Heavy), respectively (*Figure 1E*), consistent with the polysomal profile (*Figure 1B*). The overall mRNA levels were reduced by ~15% after the treatment (*Figure 1F*), which most likely reflect the coupling between translation and mRNA stability (*Radhakrishnan and Green, 2016*). Next, we quantified the translational response of the TOP mRNAs (CYYYY sequence is in the promoter

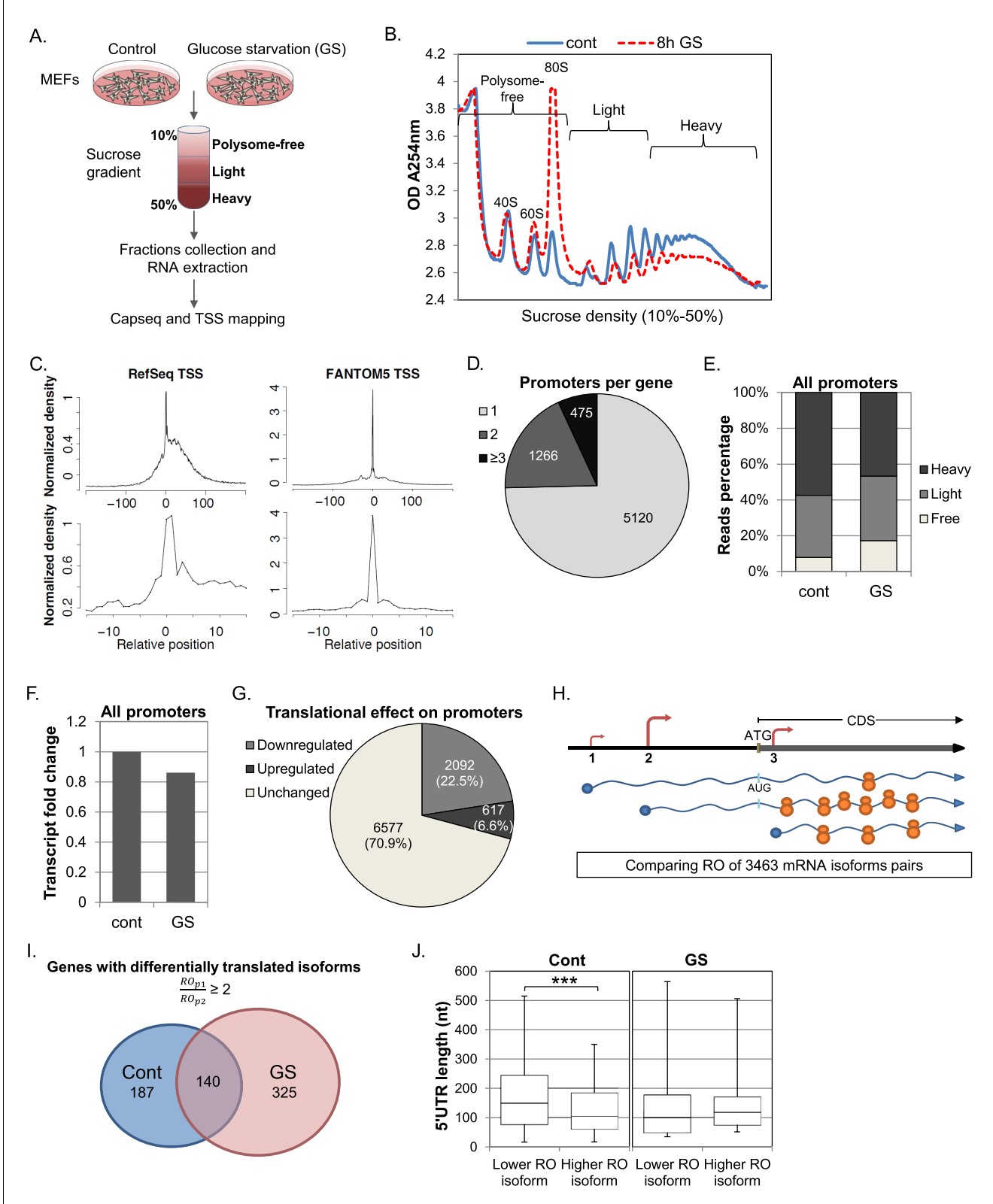

**Figure 1.** Experimental design and general analysis of the impact of TSS selection on translation efficiency. (**A**) A schematic flowchart of the biological experiment and sample preparation for the CapSeq analysis. (**B**) Polysomal profiling of MEF cells subjected to glucose starvation (GS) for 8 hr (dashed red) or untreated (blue). (**C**) Metagene analysis of CapSeq reads relative to the annotated TSS of Refseq and the summit of FANTOM5 TSSs at low and high resolutions. Only TSS regions (−200..200) with at least ten bases covered by reads were considered, and the coverage in each region was

*Figure 1 continued on next page*

*Figure 1 continued*

normalized to mean zero and standard deviation of one. The normalized coverage was then summed across all regions. (D) The relative distribution of genes with the indicated number of promoters per gene. (E) The relative global sum of reads for all promoters (9286 promoters with >500 reads) in the polysome-free, light and heavy polysomal fractions in basal (cont) and GS conditions. The presented data are the mean of two independent replicates. (F) The fold change of the global mRNA levels between the basal (cont) and GS conditions. The presented data are the mean of two independent replicates. (G) The number and percentage of promoters translationally affected by GS. Promoters that had ribosome occupancy (RO) change of two-fold or more in both repeats were considered affected. (H) A scheme demonstrating the differential translation and transcription of transcripts with alternative TSSs/promoters. The TSSs are shown as arrows and the size of the arrow denotes the strength of the TSS relative to other TSSs of the same gene. The number of ribosomes occupying each mRNA represents the extent of its translation. (I) Promoters from the same gene were paired and their ROs were compared. Pairs of promoters that had an RO difference of two-fold or more in control or GS conditions in both repeats, independently, were considered as differentially translated promoters. The numbers of genes with at least one pair of differentially translated isoforms in control and GS conditions in both repeats are presented in a Venn diagram. (J) Boxplot presentation of the distributions of the 5'UTR lengths of differentially translated isoforms in each promoter pair (as presented in I) in control and GS conditions. The bottom and the top whiskers represent 5% and 95% of the distribution, respectively.

The following figure supplement is available for figure 1:

**Figure supplement 1.** The response 5'TOP promoters to GS.

summit), since these genes are known to be particularly sensitive to stresses. As expected, TOP mRNAs were preferentially inhibited upon GS on the translational level (*Figure 1—figure supplement 1A*) while the changes of the mRNA levels were comparable with those of other mRNAs (*Figure 1—figure supplement 1B*). These data assured us in the ability of our approach to quantitatively characterize the translational properties of individual TSSs.

We next analyzed the translational response of individual promoters following the GS stress. For this we determined the ribosome occupancy (RO, the ratio between the reads in the Heavy+Light polysomal fractions to the polysome-free fraction) of transcripts emanating from each promoter and calculated the effect of GS on the RO (referred to as the RO effect) for each experiment. A promoter was considered as translationally affected if its RO was affected by at least two-fold in each of the two biological repeats (*Supplementary file 1*). With these settings, we found that following glucose starvation, 22.5% of promoters showed a reduction of two-fold or more in translation whereas only 6.6% of promoters were induced by at least two-fold (*Figure 1G*).

Our approach enabled us to compare the translation efficiency of transcript isoforms emanating from APs of the same gene as schematically shown in *Figure 1H*. In genes having multiple promoters, we separately considered each pair of promoters with sufficient reads. In basal conditions 495 such pairs from 327 genes differed in their RO by two-fold or more (*Figure 1I*). Of these, the mRNAs of 85.5% of promoter-pairs likely give rise to the same annotated ORF and the rest (14.5%) are predicted to result in proteins with alternative N-termini. When the same analysis was performed following GS, 672 pairs of promoters from 465 genes displayed RO difference of two-fold or more, resulting in 42% increase in differential translation of APs (*Supplementary file 1*). Interestingly 70% of these differentially translated pairs do not overlap with those in basal conditions (*Figure 1I*). These findings suggest that the increase in differential translation of transcript isoforms is part of the cellular stress response.

Among APs corresponding to the same ORF, those resulting in shorter 5'UTRs had higher RO under basal conditions (*Figure 1J*). On the other hand, no significant difference in 5'UTR length was seen between the more and the less efficiently translated pairs following stress (*Figure 1J*). It therefore appears that the translation regulatory features following stress differ from those operating in basal conditions.

We also compared the mRNA levels of the transcript isoforms following GS stress. We calculated the change in overall mRNA levels of each promoter following GS, and compared the effects on the APs of each gene (*Supplementary file 1*). 140 genes had at least one AP pair that showed ≥2 fold change in mRNA levels in response to stress in both experiments. Of the promoters preferentially induced following stress, 53% encoded a transcript with the same predicted ORF but shorter 5'UTR while the others were split between transcripts with the same ORF and longer 5'UTR and transcripts with different predicted ORF starts (*Figure 1—figure supplement 1C*).

## The nucleotide context of the TSS plays a role in the translational response to GS

The CapSeq data can provide insights into the impact of the exact nucleotide context of the TSS on translation efficiency. To determine the effect of the initiating nucleotides on translation efficiency we first analyzed all the non-redundant TSS positions in our data. We found that in MEFs, under basal and GS conditions, ~34% of TSSs initiated with adenosine (A), ~23% with cytidine (C), ~30% with guanosine (G) and ~13% with thymidine (T) (*Figure 2—figure supplement 1A*), similar to previous reports (*Carninci et al., 2006*; *Yamashita et al., 2006*; *Forrest et al., 2014*). Under basal conditions no significant difference was found in the RO of transcripts that vary in their first nucleotide (*Figure 2A*). In contrast, the identity of the first base was correlated with significant differences in the response to GS (*Figure 2B*). The RO of transcripts initiating with pyrimidines (C and U) was significantly reduced while those initiating with purines (A and G) appeared refractory to the stress. We performed the same analysis for initiating trinucleotides and found that under basal conditions, the RO of most trinucleotides is similar (*Figure 2—figure supplement 1B*). Notable exceptions are UGA, UGC, UGG and UGU that display higher RO and the pyrimidine-rich trinucleotides CCT, CTC, CTT, that correspond to somewhat lower RO (*Figure 2—figure supplement 1B*). The latter trinucleotides are most likely part of the TOP (CYYYY) element that was shown to confer slightly reduced translation even under optimal growth conditions (*Patursky-Polischuk et al., 2009*). Interestingly following GS (*Figure 2C*) the majority the trinucleotides with initiating pyrimidines showed reduced translation, while those with purines were largely unaffected (*Figure 2C*). Among the inhibitory trinucleotides were those that are part of the TOP element such as CCT and CTT, but also others that deviate from the TOP such as CAA, CAT, CTG, TAC etc.

Representative examples of TSSs that are separated by just a few bases within the same promoter and display differential translational reaction to GS are shown in *Figure 3*. The overall translation and mRNA levels of Atp5a1 (ATP Synthase, mitochondrial) are downregulated upon GS (*Figure 3A*, upper right panel). This promoter has several strong TSSs, three of which are adjacent to each other and designated 1, 2 and 3. Translation of TSS#1 that begins with CAT was much more strongly inhibited than that of the nearby TSSs that begin with ATT and TTT, regardless of the effect on transcription.

Another example is the p1 promoter of the Sars (Seryl-tRNA synthetase) gene that has multiple TSSs (*Figure 3B*, upper panel shows the major two). The overall translation of Sars was downregulated after GS (*Figure 3B*, upper right panel). The two major TSSs, separated only by 5nt, had a clearly differential response to GS. While in basal conditions TSS#1 and TSS#2 showed the same pattern of ribosome occupancy, following GS the translation of TSS#2 that initiates with CTC, was downregulated to a greater extent than TSS#1 that initiates with ACA. The overall translational response to GS of Sars p1 promoter was intermediate between the responses of each TSS. These examples clearly show that in addition to differential promoter regulation, the initiating sequence of an mRNA is also important for the translation efficiency of the transcript.

## The TSS nucleotides influence the cap-binding affinity of eIF4E

To gain insight into the underlying basis for the effect of cap-proximal nucleotides on translation, we analyzed eIF4E, the cap binding subunit of eIF4F. eIF4E has a single promoter that drives high level of transcription and translation under basal conditions (*Figure 4A and B*). Following GS eIF4E expression is substantially downregulated at the mRNA and translation levels (*Figure 4A and B*), which leads to reduction in eIF4E protein levels (*Figure 4C*).

A possible link between eIF4E levels and the effect of cap-proximal nucleotides under stress may involve differential binding affinity. We therefore analyzed eIF4E binding to RNAs with different first nucleotides. Most previous studies of eIF4E binding affinity used cap analogs such as m7GTP or m7GpppG and reported Kd values ranging from low μM to few nM (*Ueda et al., 1991*; *Carberry et al., 1992*; *Minich et al., 1994*; *Sha et al., 1995*; *Hagedorn et al., 1997*; *Miyoshi et al., 1999*; *Wieczorek et al., 1999*; *Hsu et al., 2000*; *von Der Haar et al., 2000*; *Niedzwiecka et al., 2002*; *Scheper et al., 2002*; *Ghosh et al., 2008*; *Thillier et al., 2012*). We chemically synthesized three capped RNA oligos as previously described (*Lavergne et al., 2008*; *Thillier et al., 2012*), 10 nt long each that differ in their first nucleotide. Human eIF4E was expressed as His-tag fusion in E. coli and purified from the soluble fraction by nickel agarose beads and subsequent gel filtration

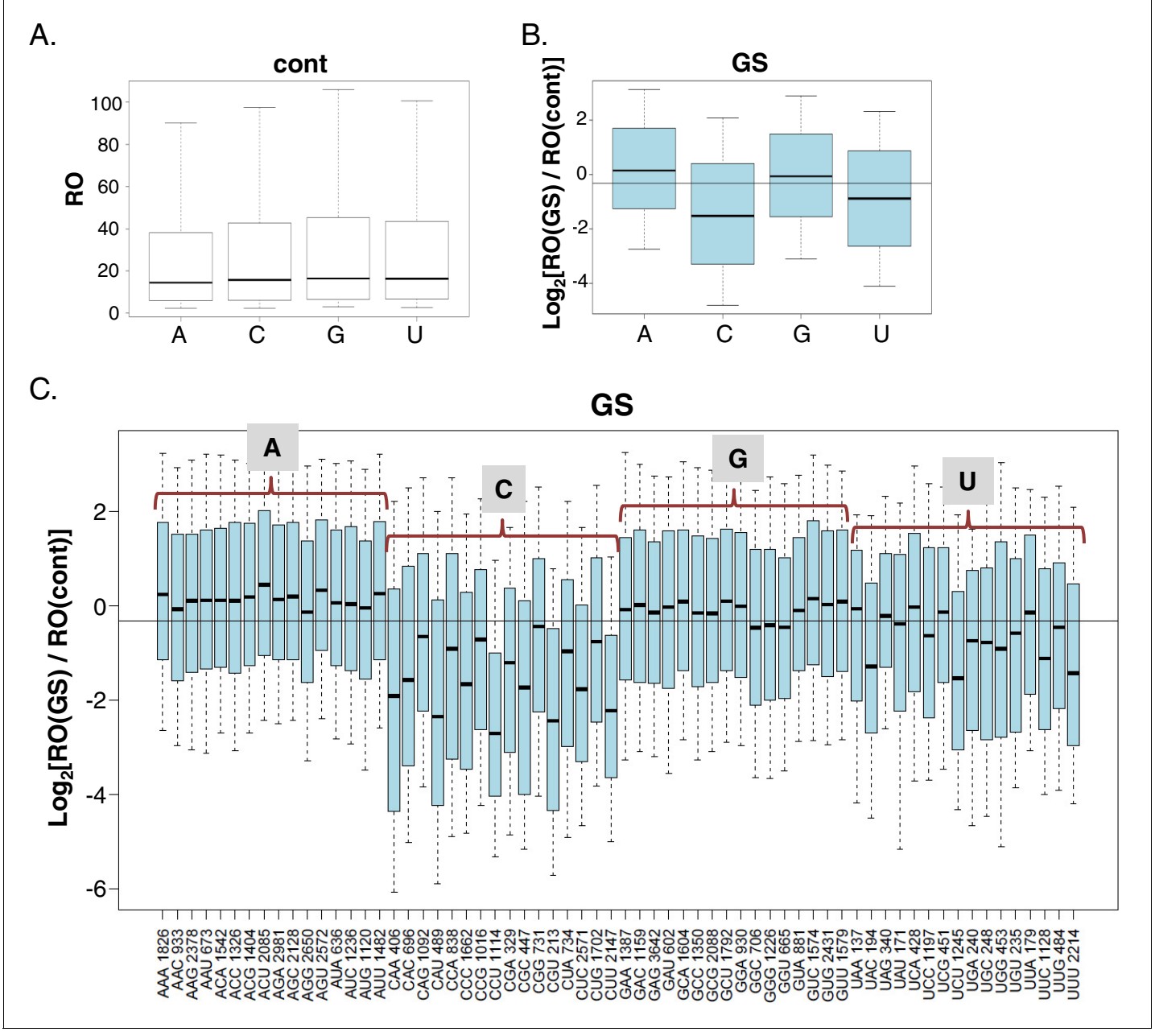

**Figure 2.** The impact of the TSS nucleotides on mRNA translation. (A) The RO distributions of transcripts that initiate with the indicated nucleotide in basal conditions. (B) The distribution of the RO effect (log transformed) of transcripts that initiate with the indicated nucleotide. The horizontal line indicates the overall median value. (C) The effect of GS on the RO for each initiating trinucleotide. The horizontal line indicates the overall median value. All the data presented in this figure are the mean of the two independent replicates. The bottom and the top whiskers represent 5% and 95% of the distribution, respectively. The number of TSSs starting with the indicated trinucleotide is indicated near each of the trinucleotide sequence.

The following figure supplement is available for figure 2:

**Figure supplement 1.** The frequency of the initiating nucleotides and their impact on basal translation.

(*Figure 4—figure supplement 1A*). Using the purified eIF4E we performed intrinsic fluorescence intensity measurements in the presence of increasing concentrations of either m7GpppG cap analog or the capped RNA oligos whose first nucleotide is C, A or G. The titration data of the RNA oligos and the cap analog are shown in *Figure 4D*. The measured Kd for the m7GpppG was 561 nM

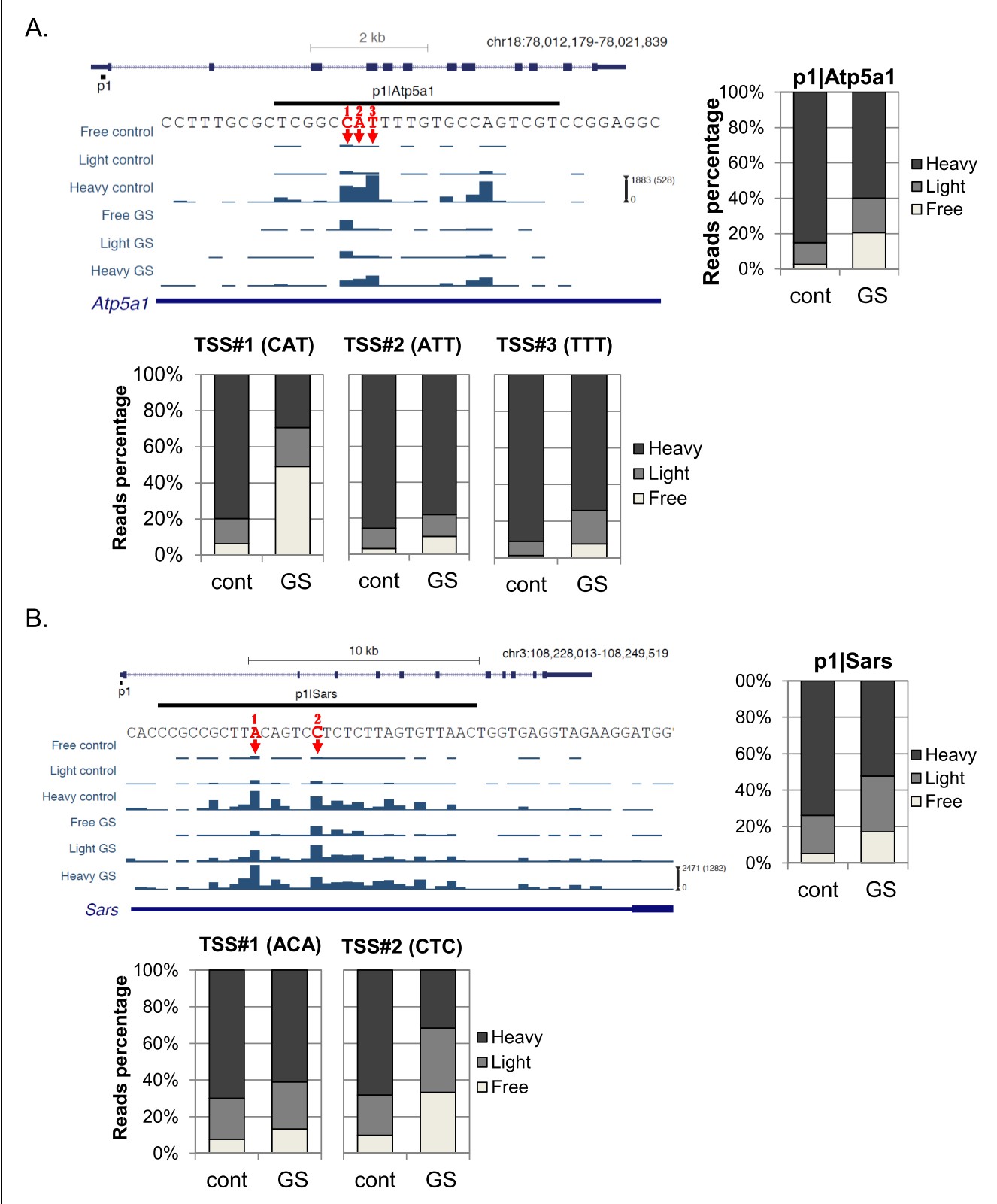

**Figure 3.** Examples of the effect of the first nucleotide on translation efficiency. (A, B) Upper left panel – The chromosomal location, genomic structure and CapSeq data of the indicated genes in each fraction in control and GS conditions. The scale of the normalized and row reads (in parentheses) is shown. The nucleotide sequence of the major alternative TSSs within the promoter are marked by arrows and numbered. The upper right panel shows

*Figure 3 continued on next page*

*Figure 3 continued*

the relative levels of all reads for the indicated promoter and the lower panels for the designated specific TSSs in basal (cont) and GS conditions. The presented data are the mean of the two independent replicates.

(*Figure 4D*). Interestingly, under the same conditions, the Kd of all capped RNA oligos was substantially lower, indicating that the RNA moiety increases the binding affinity (*Figure 4D*). Furthermore, significant differences in the Kd values between the RNA oligos were observed (*Figure 4D*, p-values are shown in *Figure 4—figure supplement 1B*). The oligo with polypyrimidine CCU as initiating nucleotides display lower affinity compared to oligos with ACU and GCU as first nucleotides. These Kd values of the mRNAs are correlated with the data of the relationship of the first nucleotides and the RO response to GS (*Figure 2C*). These findings suggest that eIF4E binding affinity contributes to the level of translation, in particular when its intracellular concentrations become limiting as in GS (*Figure 4C*). To gain further support to this idea we downregulated the levels of eIF4E using siRNA and introduced into these cells GFP reporter genes driven either by Rpl18 or CMV promoters that direct transcription start site at a C (this study) or an A (*Elfakess and Dikstein, 2008*) nucleotide, respectively. Both, the C and the A initiated transcripts were downregulated but the effect is substantially more pronounced for the C-initiated transcript (*Figure 4E*). We also analyzed these GFP reporter genes in cells treated with low concentrations of 4EGI-1, a drug that inhibits eIF4E by disrupting its interaction with eIF4G1 (*Moerke et al., 2007*). Here again the C-initiating transcript was much more vulnerable to this drug compared to the A-initiated one (*Figure 4F*), consistent with the notion that a C as an initiating nucleotide is highly responsive to fluctuations in eIF4E availability.

## Differential translation of transcript isoforms is part of the stress response

Next we looked at several genes with more than one promoter displaying differential translation and transcription. The global inhibition of translation in response to stress is mediated by the 5'TOP element that is present in almost all ribosome subunits and many of the translation initiation factors. An exception is eIF4A, the helicase subunit of eIF4F. As seen in the mapped CapSeq reads (*Figure 5A*), four promoters (p1, p2, p3 and p5) account for most of eIF4A mRNAs. p1 and p2 are positioned upstream of the annotated eIF4A protein start codon and in basal conditions these are the major promoters producing highly transcribed and efficiently translated mRNAs (*Figure 5B and C*). On the other hand, p3 and p5 promoters are intronic and positioned downstream of the annotated protein start codon so that transcription from these promoters generates isoforms with different 5'UTRs and an alternative truncated N-terminus. Upon GS p1 and p2 translation was slightly reduced (*Figure 5B*), whereas the intronic p3 and p5 promoters were significantly upregulated transcriptionally and translationally (*Figure 5B and C*). While p3 and p5 constitute a small fraction of the overall eIF4A under basal conditions, they account for more than half of eIF4A transcripts upon GS.

To investigate further the induction of the intronic promoters and their function we first confirmed with 5'RACE that p3 is indeed strongly induced by GS (*Figure 5D*) and is differentially translated compared to p1 as evident from their distribution in sucrose gradient fractions (*Figure 5E*). The putative 5'UTR of the intronic isoform contains multiple AUGs possibly generating short upstream ORFs. To gain insight into the translation initiation site of the induced protein isoform we analyzed eIF4A protein using C-terminus specific antibody. We observed an energy-stress induced polypeptide of 33 kDa that is ~13 kDa shorter than the main protein (*Figure 5F*). This truncated protein can be synthesized either from Met 121 or Met 127 of the main ORF. We used a GFP reporter that is preceded by the entire novel eIF4A 5'UTR starting from p3 up to Met121. This construct drove the expression of a single polypeptide corresponding to translation initiation at Met121, which was also induced following GS (*Figure 5G*). To examine the function of this truncated protein we constructed an eIF4A expression plasmid lacking the first 120 amino acids (eIF4AΔN). This truncation is expected to impair the helicase activity of eIF4A but to retain its interaction with eIF4G1 (*Dominguez et al., 2001*). Cells were transfected with either full-length (WT) or N-terminally truncated eIF4A together with a GFP reporter gene preceded by a 5'UTR bearing a cap-proximal secondary structure (*Figure 5H*). While the WT eIF4A has no effect on GFP expression, eIF4AΔN inhibited GFP protein levels in a dose dependent manner, indicating that eIF4AΔN acts as an inhibitor of eIF4A. Thus AP

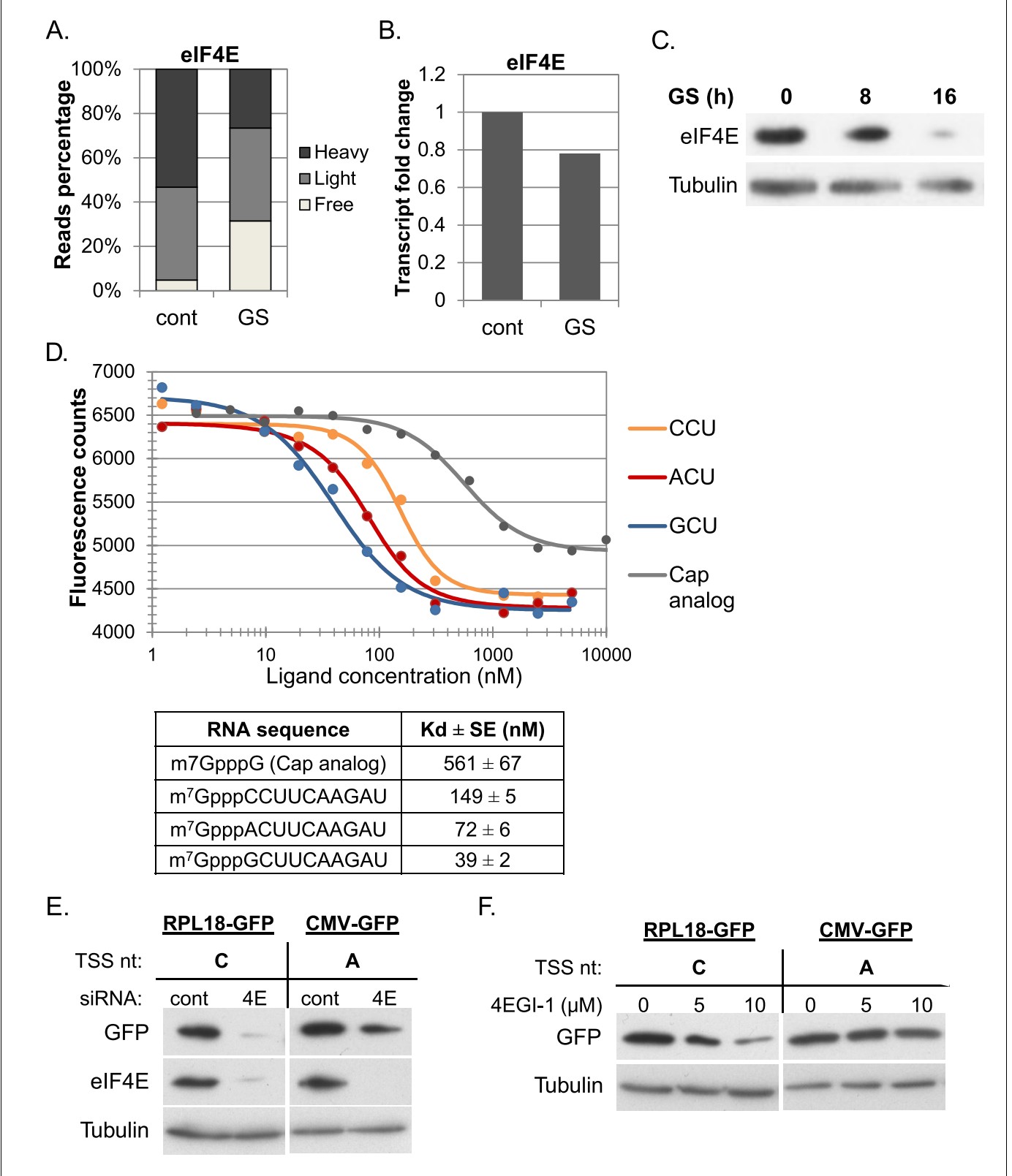

**Figure 4.** The effect of cap-proximal nucleotides on eIF4E binding affinity and activity. (**A**) The relative level of eIF4E reads (mean of two independent replicates) in the indicated fractions in basal (cont) and GS conditions. (**B**) The relative mRNA levels (mean of two independent replicates) of eIF4E in basal (cont) and GS conditions. (**C**) Representative immunoblot of total lysate using eIF4E and Tubulin antibodies following GS of MEFs for the indicated times. (**D**) Upper panel – the change in the intrinsic fluorescence of eIF4E (300 nM) in response to increasing concentrations of capped RNA

*Figure 4 continued on next page*

*Figure 4 continued*

ligands (1.25 nM–5 µM) or cap analog (1.25 nM–10 µM). The graphs represent the mean of three independent experiments with two different protein preparations. The bottom panel shows the calculated dissociation constant values (Kd) of eIF4E binding affinity to the indicated RNA ligands. (E) The effect of eIF4E knockdown on GFP expression driven by mRNA with C or A as the initiating nucleotides. HEK293T cells were transfected with either eIF4E siRNA or a non-targeting siRNA (10 nM). 48 hr later cells were transfected again with GFP reporter genes driven either by RPL18 (C) or CMV (A) promoter. Cells were harvested 24 hr after the second transfection and analyzed by western blot with GFP, eIF4E and Tubulin antibodies as indicated. (F) HEK293T cells were transfected RPL18 and CMV driven GFP reporter gene. Six hours later, increasing amounts of 4EGI-1 were added to the media as indicated. Cells were harvested 24 hr after transfection and subjected to western blot using GFP and Tubulin antibodies.

The following figure supplement is available for figure 4:

**Figure supplement 1.** Complementary data for the analysis of eIF4E binding affinity.

usage in eIF4A diminishes its activity following GS, reminiscent of the down regulation of other initiation factors following stress.

Poly(A) binding protein (Pabp), a central translation initiation regulatory factor, has several annotated promoters. The major promoter under basal conditions is p1, which generates a TOP mRNA (*Figure 6A*). This mRNA isoform is expressed 40-folds higher than the next most transcribed isoform, driven by promoter p2 (*Figure 6C*). After GS, p1 mRNA levels and translation were both strongly reduced (*Figure 6B and C*), while p2 transcription was elevated by eight-fold (*Figure 6C*) and the vast majority of p2 transcripts were heavily translated (*Figure 6B*, right). In Pabp, both p1 and p2 are positioned upstream of the annotated protein start codon, hence the differential regulation of the APs upon GS changes solely the 5'UTR length (from 467 nt to just 63 nt). Interestingly, it was previously reported by several groups that the 5'UTR of the Pabp major isoform consists of conserved A-rich sequences (ARSs) which serve as binding sites of Pabp, resulting in translational repression (*Bag and Wu, 1996*; *Hornstein et al., 1999*; *Kini et al., 2016*). These ARSs are missing in the shorter GS-induced mRNA isoform. Using 5'RACE we validated the induction of p2 mRNA isoform upon GS (*Figure 6D*) and the differential translational response of p1 and p2 to GS (*Figure 6E*). Analysis of Pabp protein levels revealed that despite the strong translational inhibition of the primary isoform its protein levels remained stable, suggesting that the highly translated GS-induced transcript isoform compensates for the diminished translation of transcripts produced from p1 (*Figure 6F*). Pabp plays an important role in translation by promoting RNA circularization mediated by interaction with the cap complex eIF4F (*Wells et al., 1998*; *Kaye et al., 2009*; *Yanagiya et al., 2009*; *Park et al., 2011*). However, tight control of Pabp levels is critical as excess of Pabp is inhibitory for translation in vivo and in vitro (*Yanagiya et al., 2010*). As eIF4F activity and the overall mRNA levels (dictating the overall abundance of poly-A tails) are downregulated in response to GS, the stoichiometry of Pabp relative to eIF4F and to the mRNA poly-A tails is elevated upon the stress and thus contributes to translation inhibition.

We examined the potential of p2 upstream region, which also serves as the 5'UTR of the major Pabp transcript (p1), to act as a promoter. The upstream sequences of p2 were placed upstream to a promoter-less *Renilla* luciferase (RL) gene and transfected into MEFs (*Figure 6G*). The results revealed that progressive addition of upstream sequences up to 282 nt relative to the initiating ATG, substantially enhanced RL activity. Further extension of p2 upstream sequences up to 456 nt resulted in diminished RL activity. The results confirm that the p2 promoter overlaps the 5'UTR of the major Pabp transcript and uncovered positive and negative regulatory elements.

Arhgef2 (Rho/Rac guanine nucleotide exchange factor 2) is another interesting example of a gene with APs that are differentially regulated by GS (*Figure 6—figure supplement 1A*, showing the five strongest promoters of Arhgef2). The p1 and p3 promoters give rise to the longest Arhgef2 isoforms and their mRNA levels are slightly inhibited following GS. p2 directs the synthesis of the shortest Arhgef2 isoform with distinct 5'UTR and N-terminus and its translation and mRNA levels are moderately upregulated. p4 and p10 also generates an alternative N-terminus isoform with different 5'UTR as well. Following GS, their mRNA levels were induced by an average of 15-folds and their translation was also induced. Thus the GS stress shifted Arhgef2 protein synthesis from one N-terminal isoform to another.

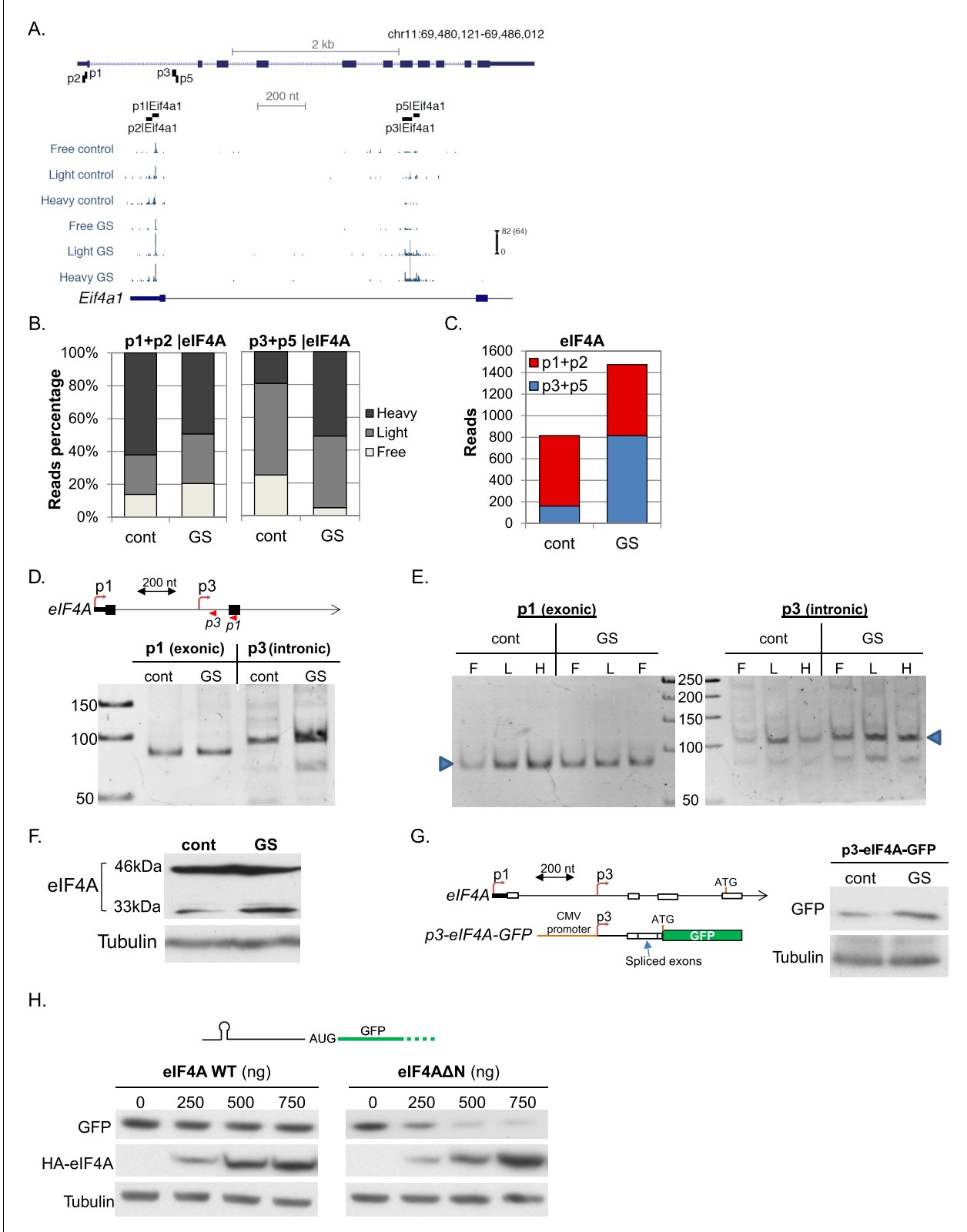

**Figure 5.** Characterization of the GS-induced AP of eIF4A. (**A**) The chromosomal location, genomic structure and CapSeq data of eIF4A in each fraction in basal (cont) and GS conditions (uniquely mapped reads). The scale of the normalized and row reads (in parentheses) is shown. The positions of the FANTOM5 promoters are also indicated. (**B**) The relative levels of the reads from the indicated promoters (mean of two independent replicates) in the three polysomal fractions in control and GS conditions. (**C**) The number of CapSeq reads of the indicated promoters in control and GS conditions

*Figure 5 continued on next page*

*Figure 5 continued*

(mean of two independent replicates). (**D**) Analysis of the p1 and p3 promoters of eIF4A by 5'RACE. Upper panel shows schematic presentation of the relevant eIF4A genomic region, the positions of the analyzed promoters and the 5'RACE reverse primers (shown by arrowheads). The lower panel is the analysis of the 5'RACE PCR products by 6% PAGE. (**E**) Analysis of p1 and p3 transcript isoforms levels by 5'RACE in the indicated fractions of the polysome profile of control and glucose starved cells. (**F**) Representative immunoblot of total lysate of MEFs with eIF4A (C-terminal epitope) and Tubulin antibodies in control and GS. (**G**) The 5'UTR of the eIF4A from the p3 TSS to Met121 was cloned downstream of the CMV promoter and upstream of the GFP reporter gene as shown in the scheme. This construct was transfected into MEFs, which were then subjected to GS. Expression of GFP was monitored by western blot with GFP antibody. (**H**) Representative immunoblot of HEK293T cells that were co-transfected with GFP reporter having hairpin structure within the 5'UTR together with increasing amounts of WT or ΔN-eIF4A as indicated.

An intriguing example of alternative TSSs selection within the same promoter following stress is shown in *Figure 6—figure supplement 1B*. Hcfc2 (Host cell factor C2) has one major expressed promoter (p1) in MEFs. Upon GS, the p1 promoter mRNA contribution increased by 1.7-folds alongside a mild increase in ribosome occupancy (*Figure 6—figure supplement 1B*, bottom panel). Interestingly, one of the weak TSSs within the Hcfc2 p1 promoter rose dramatically following GS (TSS #2). This TSS is positioned only three nucleotides upstream to the only annotated Hcfc2 protein's start codon, creating 5'UTR of only 3nt. Although Hcfc2 has a TISU (Translation Initiator of Short 5'UTR) element surrounding its initiating AUG (*Elfakess and Dikstein, 2008*; *Elfakess et al., 2011*), a 5'UTR of 3nt is too short even for TISU mediated translation initiation. Hence, the translation of the GS-induced TSS #2 is expected to start at a downstream in-frame AUG, creating N-terminally truncated isoform (34 aa shorter).

Several additional interesting examples of AP-mediated response affecting both transcription and translation are described in *Figure 6—figure supplement 2*.

## Discussion

In the present study, we provide a global view and mechanistic insights of the impact of TSS selection on translation following metabolic energy stress. Our findings uncover the critical importance of the exact TSS/cap-proximal nucleotides in the translational response to energy stress. In basal condition, the initiating nucleotides have no significant effect while the 5'UTR length appears to be an important determinant that governs the differential translation efficiency of transcript isoforms, with longer 5'UTRs diminishing translation efficiency. The exact opposite is seen during energy stress, as the effect of 5'UTR length was insignificant while the initiating nucleotides appear to be responsible, at least in part, for the difference in the RO effect. This phenomenon was clearly evident from cases in which adjacent nucleotides from the same promoter display dramatic differences in the translational response to the stress. In these instances the 5'UTR length and sequence are almost identical. The differential sensitivity to the stress is particularly apparent between the purine and pyrimidine nucleotides. Although the first nucleotide contributed greatly to this differential effect, the following nucleotides are also important. The nucleotides that confer inhibition of translation following the stress start in most cases with cytidine, as does the TOP element. Intriguingly, the following nucleotides in GS-repressed trinucleotides do not always follow the TOP consensus of uninterrupted stretch of pyrimidines. Starting purines seem to confer greater resistance to the inhibition of translation. While the reporter gene assays clearly show the importance of the first nucleotide and isoforms generated from AP for differential translation (*Figures 4E, F* and *5G*), it is possible that other features in endogenous transcripts may contribute to translation. Furthermore eIF4G1 and eIF4A, the partners of eIF4E in eIF4F, may also contribute to the differential affinity via their RNA binding domains. Thus, our findings expand the repertoire of regulatory sequences that mediate the effect of the stress on translation.

We provide a potential mechanistic link between eIF4E and the effect of TSS nucleotides on translation following GS. First, eIF4E translation and protein levels are downregulated. This is in addition to the inhibition of its activity by 4EBP upon this stress (*Bolster et al., 2002*; *Dubbelhuis and Meijer, 2002*; *Krause et al., 2002*; *Reiter et al., 2005*). The translation inhibition of eIF4E may be a backup mechanism for 4EBP-mediated inhibition. Second, mRNAs that differ in their cap-proximal nucleotides display differential affinity towards eIF4E in correlation with the observed

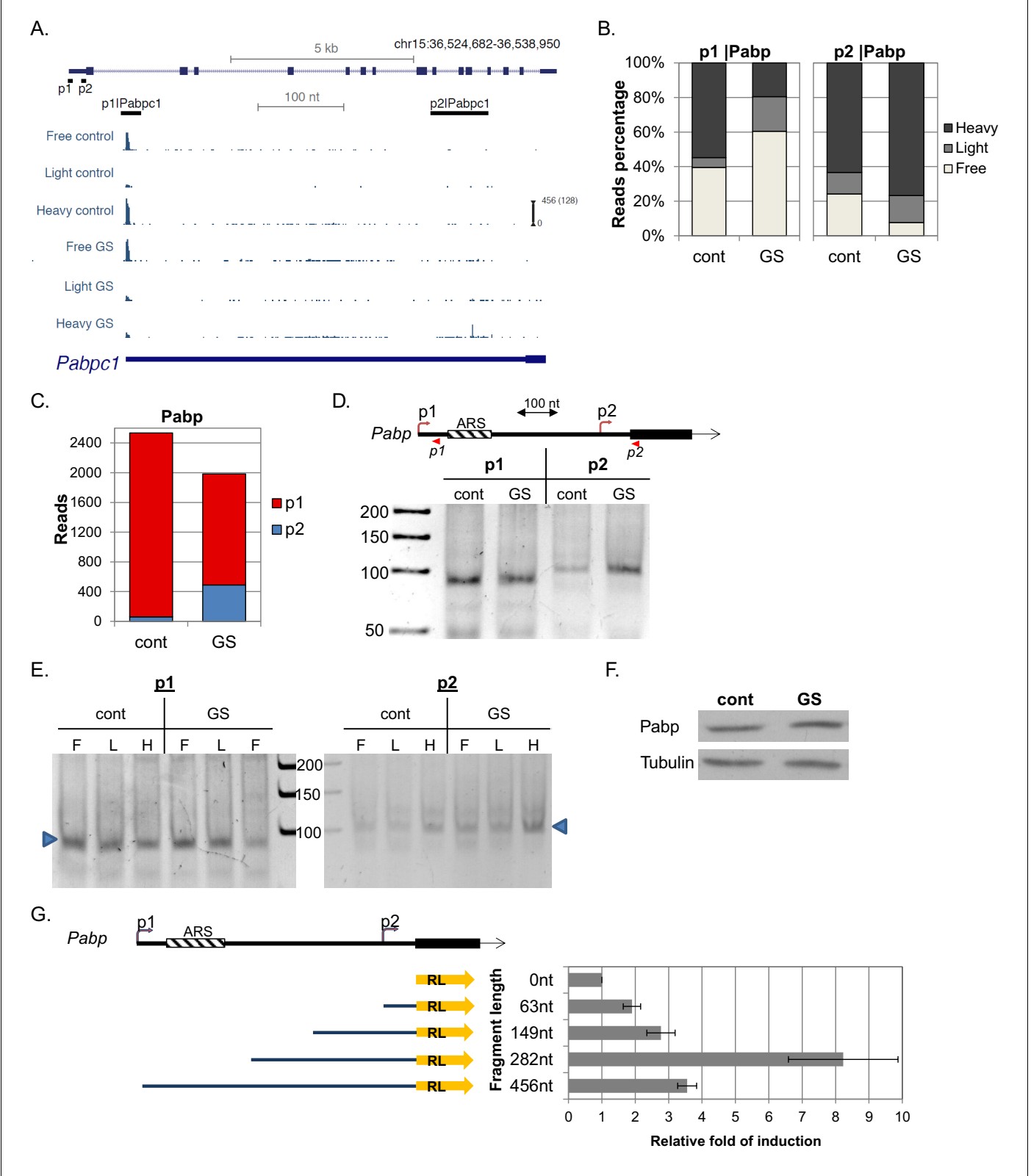

**Figure 6.** Characterization of Pabp APs. (**A**) The chromosomal location, genomic structure and CapSeq data of Pabp in each fraction in basal (cont) and GS conditions. The scale of the normalized and row reads (in parentheses) is shown. The positions of the FANTOM5 promoters are also shown. (**B**) The relative levels of the indicated promoter reads of the polysomal fractions in control and GS conditions (mean of two independent replicates). (**C**) The total CapSeq reads of the indicated promoters in control and GS conditions (mean of two independent replicates). (**D**) 5'RACE analysis of p1 and

*Figure 6 continued on next page*

*Figure 6 continued*

p2 promoters of Pabp. Upper panel shows a schematic presentation of Pabp region containing the p1 and p2 promoters, their positions, 5'RACE reverse primers (shown as arrowheads) and adenine-rich autoregulatory sequences (ARSs). The lower panel is the analysis of the 5'RACE PCR products by 6% PAGE. (E) Analysis of p1 and p2 transcript isoforms levels by 5'RACE in the indicated fractions of the polysome profile of control and glucose starved cells. (F) Representative immunoblot of total lysate of MEFs with anti-Pabp and anti-Tubulin in control and GS conditions. (G) Characterization of Pabp p2 upstream region as a promoter. A scheme of p2 regulatory sequences of the indicated lengths (starting from the AUG) that were cloned upstream to a promoter-less *Renilla* luciferase (RL) reporter gene is shown on the left. These constructs were transfected into MEFs together *Firefly* luciferase reporter gene that served as an internal control. *Renilla* and *Firefly* luciferase activities were measured 24 hr after transfection. The results represent average ± SE of 4 transfection experiments.

The following figure supplements are available for figure 6:

**Figure supplement 1.** Examples of differential transcription and translation response to GS of APs.

**Figure supplement 2.** Additional examples of differential transcriptional and translational response to GS of APs.

sensitivity to the stress. Specifically, the TOP-like CCU that is highly repressed following GS exhibits much lower affinity compared to RNAs that initiate with an A or G which we found to be more resistant in their translation to the stress. Several previous studies that addressed the regulatory mechanism of the TOP element uncovered several trans-acting factors, positive and negative, that mediate the translational control of these mRNAs (*Meyuhas and Kahan, 2015*). Our findings add an additional layer of regulation of these mRNAs by demonstrating that under energy stress conditions, their inhibition/resistance is, at least in part, the outcome of the intrinsic properties of the translation machinery. It is possible that the signal-modulated trans-acting factors of the TOP element act to modify the eIF4E-mRNA affinity. For instance LARP1 interacts with both the TOP element and PABP to stimulate TOP mRNA translation (*Tcherkezian et al., 2014*). As an RNA binding protein with preference to pyrimidine-rich sequences this factor can greatly increase the recruitment of eIF4E to these mRNAs via PABP, which also interacts with eIF4G1.

The involvement of the binding affinity of TOP mRNAs to eIF4E was previously studied by Shama et al using in vitro translation assays of TOP and non-TOP mRNAs in the presence of the cap analog m7GpppG (*Shama et al., 1995*). As the two types of mRNA were inhibited to a similar extent by the cap analog it was concluded that the TOP and non-TOP mRNAs have similar eIF4E binding affinity. The discrepancy between this study and our results can be explained by the use of different assays. In the present study the binding affinity was measured directly using known amounts of RNA and eIF4E protein. Shama et al. inferred the relative affinity from an indirect assay in which the actual amount of eIF4E and the mRNAs are not known and the concentration of the cap analog was very high, above the saturating range. Another study does suggest a role of eIF4E in mTOR-regulated TOP mRNA translation on the basis of short-term pharmacological inhibition of mTOR in 4EBP-deficient cells (*Thoreen et al., 2012*). However, this conclusion was challenged since other stresses (oxygen and amino acid deficiencies) that also diminish mTOR activity resulted in TOP mRNAs repression in these cells (*Miloslavski et al., 2014*). It is possible that the reduction in eIF4E availability by means other than 4EBP may contribute to this regulation.

We also uncovered hundreds of genes in which their APs were differentially translated. Remarkably, this phenomenon was particularly apparent following GS as the number of genes with 5'end isoforms displaying differential RO was substantially elevated. The detailed analyses of several intriguing examples of transcript isoforms with differential translation reveal the potential of translational control via APs. Specifically, we found that differential transcription and translation of two central translation initiation factors, eIF4A and Pabp, contribute to the global inhibition of translation following energy stress. Both have GS-induced isoforms that result in up-regulation of inhibitory eIF4A and Pabp proteins. The GS-induced downstream intronic promoter of eIF4A drives the expression of an isoform with long 5'UTR that consists of 10 uAUGs, yet this isoform is efficiently translated under GS. One possibility is that the GS-induced translational activity involves an internal ribosome entry site (IRES) that can bypass the inhibitory effect of uAUGs. Alternatively, translation can be activated by uORF-mediated mechanisms in which phosphorylation of the subunit of eIF2α favors translation of genes containing multiple uAUGs (*Young and Wek, 2016*).

In summary, by measuring isoform-specific translation in basal and energy stress conditions we uncovered previously unknown regulatory mechanisms that broaden our understanding of how stress alters cellular translatome and transcriptome and how these processes are coordinated. We anticipate that further analysis of the data and focusing on specific examples will provide additional novel insights of transcription-translation links.

## Materials and methods

### Analysis of global translation using CapSeq

Untreated and 8 hr glucose starved MEFs were incubated with 100 µg/ml Cycloheximide (Sigma) for 5 min and then washed twice with cold buffer containing 20 mM Tris pH 8, 140 mM KCl, 5 mM MgCl2 and 100 µg/ml Cycloheximide. The cells were collected and lyzed with 500 µl of same buffer that also contains 0.5% Triton, 0.5% DOC, 1.5 mM DTT, 150 units RNAse inhibitor (Eurx) and 5 µl of protease inhibitor (Sigma). The lyzed samples were centrifuged at 12,000g at 4°C for 5 min. The cleared lysates were loaded onto 10–50% sucrose gradient and centrifuged at 41,000 RPM in a SW41 rotor for 90 min at 4°C. Gradients were fractionated and the optical density at 254 nm was continuously recorded using ISCO absorbance detector UA-6. The collected samples were then merged to create three fractions: Polysome-free (F), Light (L) and Heavy (H). The use of three pools is sensitive to large changes in translation but relatively insensitive to smaller changes. RNA was isolated for each fraction using Trizol and Direc-Zol RNA mini-prep kits (Zymo Research). RNA spikes (0.25 ng GFP and 1pg Luciferase) that were transcribed (P1300, Promega) and capped (S1407, NEB) in-vitro using T7 RNA Polymerase were added into each fraction. Equivalent RNA volume was taken from each fraction for the library preparation for TSSs sequencing using the CapSeq method (with modifications) as previously described (*Gu et al., 2012*). Specifically, the RNA samples were treated with Terminator 5´-Phosphate-Dependent Exonuclease (TER51020, Epicenter). Then, reaction volume was enlarged and CIP (M0290, NEB) was added to dephosphorylate non-mRNA RNAs (tRNA and 5S rRNA). DNaseI was added in this step. In order to remove mRNA 5'Cap, the samples were treated with TAP (T19050, Epicenter) and then RNA linker was ligated at the 5'end of the formerly capped mRNA by T4 RNA ligase (M0204, NEB). cDNA was prepared from each RNA sample by using hexamer random primers linked to illumina 3' Rd2 seq primer, index (unique barcode for each sample), 4nt-long UMI and P7 illumina adaptor. To increase the cDNA quantity, a linear PCR amplification (second strand synthesis) was performed with forward primer containing the ligated linker sequence and carries P5 illumina adaptor. Final library amplification PCR step was performed after calibrating the numbers of PCR cycles. Size selection with magnetic beads (Ampure XP, according to the manufacturer guidelines) was performed after cDNA synthesis for removal of fragments < ~150 nt and also after linear amplification and final PCR steps, in which sizes of >200 bp and <500 bp were selected. The deep sequencing was performed with HiSeq High-Throughput Sequencing System.

### Deep sequencing data mapping and analysis

Reads were mapped to the mouse genome (mm9 assembly) using STAR aligner and to the GFP and luciferase sequences using Bowtie2. The BAM file was altered to contain just the first base of the alignment using a custom Java script (*Supplementary file 2*) and coverage tracks were prepared using genomeCoverageBed. The tracks were then normalized by the numbers of reads mapping to the GFP/LUC (spikes) and the polysomal normalization factors. The numbers of reads mapping to each TSS defined by the FANTOM5 project (phases 1 + 2, permissive set) was counted using htseq-count. Log2-transformed ratios were computed after adding a pseudo-count of 0.5 to the each normalized read number. Promoters were assigned to Ensembl 75 transcript models (ENSTXXX). A promoter was considered the 'annotated' promoter of a transcript if it was within 100 nt of its 5' end and 'Exonic' if it overlapped exons of the 5'UTR of the transcript. Ribosome occupancy was defined as the ratio of Light+Heavy fractions and the polysome-free fraction. RO effect=$\frac{\text{RO}_{\text{GS}}}{\text{RO}_{\text{cont}}}$; mRNA levels change after GS=$\frac{\text{GSreads}_{(\text{free+Light+Heavy})}}{\text{Contreads}_{(\text{free+Light+Heavy})}}$. Calculation of the differences of the RO between promoter pairs is shown in *Figure 1I* and for the mRNA levels (*Figure 1—figure supplement 1C*) is as follows:

mRNA differential effect = $\frac{\text{mRNAlevelschangeafterGS(p1)}}{\text{mRNAlevelschangeafterGS(p2)}}$. TOP element promoters were defined as promoter summits initiated with CYYYY sequence.

The data can be accessed in Gene Expression Omnibus (GEO) accession no. GSE93981 or tracked using UCSC genome browser at: ftp://ftp-igor.weizmann.ac.il/pub/hubANA.txt

## Cells, antibodies, plasmids and siRNA studies

MEFs from a WT mouse (from Benois Viollet, INSERM, Paris) and HEK293T (from ATCC) were maintained in DMEM supplemented with 10% fetal calf serum, 100 units/ml penicillin, 100 mg/ml streptomycin and 1 mM Sodium pyruvate (for MEFs only). Cells were tested negative for mycoplasma by PCR. Cells were harvested after the indicated period of glucose starvation [glucose free media (11966025, Gibco) with 10% dialyzed FBS] or 24 hr after transfection and lysates were subjected to SDS-polyacrylamide gel electrophoresis followed by western blot. The levels of proteins were determined using the following antibodies: anti-GFP (ab290, Abcam), anti-eIF4A (NBP2-24632, Novus; AAS65480C, Antibody verify), anti-eIF4E (ab33766, Abcam), anti-HA (ab9110, Abcam), anti-Pabp (sc-32318, Santa Cruz) and anti-Tubulin (Sigma).

The pEGFP-N1 (Clontech), pEGFP-N1 with 5'UTR bearing secondary structure and the HA-eIF4A plasmids were previously described (*Elfakess et al., 2011*). To construct the N-terminal truncated eIF4A we used the T-PCR method (*Erijman et al., 2011*) with HA-eIF4A as a template. Sequences of eIF4A starting from p3 TSS to Met121 were amplified from MEFs genomic DNA (intronic part) and cDNA (exonic part) and cloned downstream to the CMV promoter of in the pEGFP-N1 using two steps RF-cloning. The p2 upstream sequences of Pabp were amplified from MEFs cDNA and used to replace the CMV promoter in CMV-RL construct using T-PCR. The GFP reporter under the control of the Rpl18 promoter was described (*Sinvani et al., 2015*). All the constructs were verified by sequencing.

For knocking-down eIF4E, HEK293T cells were transfected with Dharmacon siGENOME SMART pool siRNA (M-002000–00, Thermo Scientific) using DharmaFECT1 transfection reagent. The Dharmacon ON-TARGETplus Nontargeting siRNA #3 was used as a negative control. 48 hr after the initial transfection, cells were transfected with the indicated GFP reporter plasmids. Cells were harvested 24 hr after the second transfection.

## Expression and purification of eIF4e and fluorescence measurements

eIF4e was purified from E.coli BL21(DE3) bacteria transformed with eIF4E-pET-30a construct (kindly provided by Franck Martin, Université de Strasbourg, France). Bacteria were grown with Kanamycin at 37°C up to OD(600) = 0.8. Then, IPTG (0.5 mM) was added and bacteria were harvested after overnight incubation at 18°C. The samples were lysed using sonication in TPA buffer (20 mM Tris-HCl, pH7.5, 10% Glycerol, 0.1 mM EDTA, 1 mM DTT (Sigma), 100 mM KCl) added with 2.5 mM PMSF, 10 mM $\beta$-mercaptoethanol and protease inhibitor cocktail (Sigma). All the buffers were prepared using Diethylpyrocarbonate (DEPC)-treated double distilled water. The lysate was subjected to centrifugation and subsequent ultracentrifugation in order to remove membrane associated RNAses. The soluble fraction was purified on gravity column using Ni-NTA His-Bind Resin (70666, Novagen), eluted with the TPA buffer containing 250 mM imidazole and subsequently dialyzed. Fractions containing eIF4E protein underwent gel filtration chromatography (Superdex 75, GE Healthcare).

Capped RNA oligos were chemically synthesized as previously described (*Lavergne et al., 2008*; *Thillier et al., 2012*). For the label-free fluorescence measurements, purified eIF4E (300 nM) in TPA buffer with 0.1% Pluronic F-127 (NanoTemper) and 50 µg/ml yeast tRNA, was mixed with increasing concentrations of cap analog (1.25 nM to 10 µM; S1407, NEB) or capped RNA oligos (1.25 nM to 5 µM). The reaction was centrifuged at 12,000 rpm for 5 min and then premium-coated capillaries (MO-Z005, NanoTemper) were loaded with the samples, and intrinsic fluorescence measurements were performed in a Monolith NT.LabelFree instrument (NanoTemper). Data analysis was performed with the Monolith NT.Analysis software (NanoTemper).

## 5′-Rapid Amplification of cDNA Ends (5′-RACE)

Total RNA extracted from control and 8 hr glucose starved cells was used as a template to create cDNA with superscript II (Invitrogen) according to the manufacturer's instructions using gene-specific primers. The cDNA was purified using a PCR purification kit (Qiagen, Germany), followed by addition of polyG to the 5'-end using TdT enzyme (Promega) for 1 hr at 37°C. The reaction was

terminated by heat inactivation for 15 min at 65°C, and the products were purified with the PCR purification kit (Qiagen). The modified cDNA was used as a template for PCR with Phusion polymerase (NEB) using nested reverse primers and forward PolyC primer. PCR products were run on 6% polyacrylamide gel.

## Acknowledgements

We are grateful to Hadar Sinvani and Ora Haimov (our lab) for assistance, reagents and plasmids used in eIF4E experiments, Shira Albeck from the Structural Proteomics Unit (WIS) for the gel filtration of eIF4E, Yoav Lubelsky (Biological Regulation, WIS) for his help in library construction, Shlomit Gilad (G-INCPM, WIS) for the deep sequencing, Haleli Sharir (G-INCPM, WIS) for helping with the fluorescence measurements studies, Théo Guez for his technical assistance in the synthesis of capped RNA oligos, Gideon Schreiber (Biomolecular Sciences, WIS) for helpful advice and Franck Martin (Université De Strasbourg) for the His-eIF4E construct. This work was supported by grants from Israel Science Foundation (#1168/13) (RD), ISF-INCPM (2359/15) (RD) and the Minerva Foundation (#712278) (RD), Alon Fellowship (IU), European Research Council (Project 'lincSAFARI') (IU), Israeli Science Foundation (1242/14 and 1984/14) (IU), the I-CORE Program of the Planning and Budgeting Committee and The Israel Science Foundation (# 1796/12) (IU), the Minerva Foundation (IU), the Fritz-Thyssen Foundation (IU), The Abramson Family Center for Young Scientists (IU). RD is the incumbent of the Ruth and Leonard Simon Chair of Cancer Research. IU is incumbent of the Sygnet Career Development Chair for Bioinformatics.

## Additional information

### Funding

| Funder | Grant reference number | Author |
|---|---|---|
| Israel Science Foundation | 1168/13 | Rivka Dikstein |
| Minerva Foundation | 712278 | Rivka Dikstein |
| Israel Science Foundation | INCPM 2359/15 | Rivka Dikstein |
| European Research Council | lincSAFARI | Igor Ulitsky |
| Minerva Foundation | | Igor Ulitsky |
| Alon Fellowship | | Igor Ulitsky |
| Israel Science Foundation | 1242/14 | Igor Ulitsky |
| Israel Science Foundation | 1984/14 | Igor Ulitsky |
| Israeli Centers for Research Excellence | 1796/12 | Igor Ulitsky |
| Fritz Thyssen Stiftung | | Igor Ulitsky |
| The Abramson Family Center for Young Scientists | | Igor Ulitsky |

The funders had no role in study design, data collection and interpretation, or the decision to submit the work for publication.

### Author contributions

AT-B-H, Conceptualization, Data curation, Formal analysis, Investigation, Methodology, Writing—original draft, Writing—review and editing; J-JV, FD, Resources, Methodology; IU, Data curation, Software, Formal analysis, Writing—review and editing; RD, Conceptualization, Formal analysis, Funding acquisition, Investigation, Writing—original draft, Writing—review and editing

### Author ORCIDs

Igor Ulitsky, http://orcid.org/0000-0003-0555-6561
Rivka Dikstein, http://orcid.org/0000-0002-6251-4723

## Additional files

### Supplementary files

• Supplementary file 1. An Excel file containing gene lists corresponding to *Figure 1* as indicated in the name of each sheet.

• Supplementary file 2. A Java script for extraction the first base in each transcript.

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
