## [Decision Letter]

Thank you for submitting your article "TSS nucleotides via differential eIF4E binding and alternative promoter usage mediate translational response to stress" for consideration by *eLife*. Your article has been favorably evaluated by James Manley (Senior Editor) and three reviewers, one of whom, Timothy W Nilsen (Reviewer #1), is a member of our Board of Reviewing Editors. The following individual involved in review of your submission has agreed to reveal their identity: Orna Elroy-Stein (Reviewer #3).

The reviewers have discussed the reviews with one another and the Reviewing Editor has drafted this decision to help you prepare a revised submission.

As you will see all are positive about the work but some essential changes must be made via revision. In that regard please address the major comments of reviewer 2 as thoroughly as possible. Also explain CapSeq in the text and add the supplementary material requested during revision by reviewer 3.

Reviewer #1:

In this interesting manuscript authors provide new insight into translational dynamics that underlie cellular responses to energy stress.

Major findings include the demonstration that under limiting eIF4E levels, different cap proximal nucleotides have different affinities for the factor with purine containing TSS's preferentially recognized.

Additionally, the authors show that under stress a new isoform of eIF4A is synthesized that acts as an inhibitor of eIF4A and that PABP is upregulated via synthesis of isoforms with shorter 5'UTRs.

Overall, the work is well performed and carefully presented and is viewed to be appropriate with minor revision for *eLife*.

Major point the use of the term translational efficiency throughout should be changed to ribosome occupancy.

Reviewer #2:

In this manuscript, Dikstein and colleagues investigate the relationship between transcription start site usage and translational efficiency during energy stress. The authors perform a global analysis of transcription start sites (using CapSeq) across polysome fractions in the presence and absence of energy stress, with the goal of identifying changes in translational efficiency that correlate with changes in promoter usage. These data reveal a striking relationship between the identity of the cap-proximal nucleotides and the translational response to stress, with mRNAs beginning with purines being relatively refractory to stress. Biochemical analyses indicate that the cap-binding protein eIF4E has an intrinsic preference for purine-initiated capped mRNAs, which – together with the finding that eIF4E levels are reduced during stress – may explain the translational efficiency data. The authors further present data that stress-induced changes in start site usage for the eIF4A and PABP mRNAs also contribute to the stress response. Together the authors present a model in which changes in TSS selection induced by stress are a critical component of the translational response to stress, with the initiating nucleotides playing a particularly important but previously unappreciated mechanistic role.

Overall, the manuscript reports an exciting finding that would be of interest to those interested in gene regulation. However, the current version of the manuscript lacks sufficient validation of the genome-wide datasets on which the entire paper is based. In addition, there are a number of caveats to both the experimental approaches and interpretations that should be addressed. For these reasons, I would only recommend publication in *eLife* after addressing all of the major concerns listed below.

1) The authors' measure of "translational efficiency" used for all analyses is poorly chosen and can be more accurately described as a measure of "ribosome occupancy". Rather than collecting many fractions from across a sucrose gradient in order to infer the number of ribosomes per mRNA (as done in some published methods, e.g. Wang et al. Mol Sys Biol 2016), the authors only collected the "free", light and heavy polysomes. Even with these three pieces of data, the authors combined the light and heavy polysome data for unknown reasons and calculated "translational efficiency" as simply (light+heavy)/(light+heavy+free). The result is a metric that is extremely insensitive to changes in translation rate, since only if an mRNA shifts >2-fold in/out of the "free" fraction containing 0-1 ribosomes will an mRNA be labeled as down/up-regulated. This is problematic for the authors' conclusion that >70% of transcripts are unchanged at the translational level in response to stress (Figure 1) and implies that there are likely many more changes in translational efficiency than the authors capture with their method. More generally, the use of this translation metric rather than a superior alternative diminishes the impact of the manuscript. The authors should clearly explain somewhere in the manuscript that the use of three pools (rather than many small fractions) is sensitive to large changes in translation but relatively insensitive to smaller changes.

2) There is an underlying assumption of the authors' analyses that is never explicitly stated: That transcript isoforms found to differ in the location of the 5' cap are identical in all other respects. Alternatively, transcripts generated by different promoters might also be spliced differently or use alternative polyadenylation sites, any of which could contribute to the measured translational efficiencies. By attributing any differences in translation to transcription start site, the authors are implying causation from what is only a correlation. In addition to raising this possibility in the text, the authors should undertake some validation experiments for some of their example genes to show that differences in transcription start site alone are sufficient to impart different polysome sedimentation to the mRNAs.

3) Based on the examples of CapSeq reads depicted (e.g., Figure 3), the number of reads mapping to individual genes seems relatively low (though no scale is provided on the y axis). In addition, because the CapSeq method relies on ligation to the 5' ends of RNAs, sequence differences at the 5' end likely affect how efficiently ends are captured by CapSeq. For these reasons, it's important to undertake some validation of the quantitative changes in transcription start site the authors describe, by using alternative methods such as Northern blotting. In addition, the wording used to describe quantitative changes in start-site usage should be toned down from its current form to be more consistent with what the data conclusively demonstrate.

4) From Figure 3 onward, the paper reads more like a collection of cherry-picked anecdotes rather than a systematic study of a biological phenomenon. While many intriguing examples are shown in Figure 3, Figure 5 and Figure 6, there's no indication as to how widespread or representative these examples are on a transcriptome-wide level. The ending of the paper is particularly weak, as the arbitrary examples of Arhgef2 and Hcfc2 are presented without any obvious connection to the biology being explored in the manuscript. These issues should be addressed through a combination of additional analyses and textual changes.

Reviewer #3:

This elegant study address the usage of AP and TSS in response to energy stress and provides global and unbiased information about the translation status of each and every transcript. By doing so, an additional layer of complexity in gene expression is uncovered which refers to the contribution of transcription-translation linkage to proteome shaping in response to stress. This by itself is an important take-home message, which opens up the door to future similar studies concerning the transcription-translation linkage angle in response to other stressors and physiological signals. Moreover, there are additional novel findings, holding important implications. First, is the discovery that the affinity of eIF4E to capped mRNA with a proximal C nucleotide is significantly lower compared to its affinity to capped- proximal purines. The experiment was performed using cutting-edge label-free technology. This is a major contribution to the long-lasting question related to TOP-mRNAs translation regulation under conditions of limiting eIF4E. The purpose of proximal C nucleotide, *only* in TOP-mRNAs, was never realized before. Thus, the current study points out for the first time at the importance of intrinsic properties of the translation machinery to TOP mRNA translation regulation by illuminating the critical importance of TSS/cap proximal nucleotides. Second important discovery is the stress-induced differential transcription of eIF4A and PABP genes which drives the (i) formation of inhibitory eIF4A isoform and (ii) high levels of PABP via translational regulation mediated by the alternative 5'UTR. Since both genes encode translation regulatory factors, this finding provides a demonstration of the tight linkage between transcription and translation in response to stress signals. Additional insights of the study relate to interesting examples many of which are relevant to gene expression during differentiation and cellular transformation.

Comments:

The strengths of this study are several fold: (I) generation of an important high quality dataset which is useful for scientists interested in further analyses; (II) important insightful information gathered by the analysis described in the manuscript; (III) experimental validation of several high impact examples. The weakness of the manuscript in its present form is the absence of Supplementary excel file Tables containing the information related to the performed analysis (item II). Such tables, containing list of genes grouped and itemized according to specific features described and discussed in this study should be added.

Figure 1 and corresponding text in the second paragraph of the subsection “Marked increase in differential translation of alternative promoters following stress”: although the experimental steps are well explained, it is not clear how the data quantitatively characterize the translational properties of individual TSS. I guess it will be clearer to the translation field readers if the author will add an explanatory note about the reads length and what one can learn from each read. For example, the note at the beginning of the Results section explaining that ribosomal footprinting is not suitable here, is helpful for understanding. I suggest to add an additional explanatory sentence to clarify that downstream sequences shared by transcripts with different TSS cannot be found as separate reads. The fact that all reads contain the TSS itself is not trivial for people used to think about polysome-generated reads in classical ribosomal profiling experiments. A few words about the CapSeq methodology and which kind of reads to expect will help to understand it. In addition to the explanatory text in the Results section, please also add to the Methods section (end of subsection “Analysis of global translation using CapSeq”) the details of the size selection. A small illustration at the bottom of Figure 1 may enhance clarity too.

The explanatory text to Figure 1 (subsection “Marked increase in differential translation of alternative promoters following stress”, fourth paragraph) is not clear. Moreover, numbers mentioned in the text do not correspond to the numbers on the figure itself.

Legend to Figure 1 – The text in the last 4 lines is not included in figure.

Polysomes association may reflect attenuation at the elongation level. See for example Shenton et al. JBC 2006 (showing translation attenuation at the elongation level in response to oxidative stress). This notion should be mentioned in the text. In addition, please add to the legend to Figure 1): "due to translation regulation at both the initiation and elongation levels".

Informative details on the genomic locus at the lower part of Figure 5 (exon1, start codon, intron 1, exon2), as well as to Figure 6 and Figure 6—figure supplement 1, should be added.

Figure 4: p values should be part of Figure 4 (not in the supplement).

Subsection “The TSS nucleotides influence the cap-binding affinity of eIF4E”, last paragraph: should add after 'RNA oligos' "whose first three nucleotides are CCU, ACU or GCU".

Discussion, second paragraph: The work of Tcherkezian et al. 2014 is incorrectly interpreted. In this paper LARP1 was identified in a proteomic screen for proteins that associate with the mRNA 5′ cap, but was shown to be associated with PABP, not with eIF4E or eIF4G (whereas it does co-immunoprecipitated with eIF4A).

Subsection “Differential translation of transcript isoforms is part of the stress response”, second paragraph – add a reference to the statement that the truncated eIF4A retains its interaction with eIF4G1.

Discussion, end of fourth paragraph: IRES is not the only possibility. The uAUG-burdened long 5'UTR of eIF4A might confer either IRES-independent or 5'cap-dependnet translation regulation (see review by Young and Wek, JBC 2016).

Discussion, last paragraph: To be specific, must change from stressful conditions to energy stress. Indeed it may imply that a similar mechanism is operative in response to other stressful conditions, but this should be added as a possibility.

---

## [Author Response]

As you will see all are positive about the work but some essential changes must be made via revision. In that regard please address the major comments of reviewer 2 as thoroughly as possible. Also explain CapSeq in the text and add the supplementary material requested during revision by reviewer 3.

Reviewer #1:

[…] Major point the use of the term translational efficiency throughout should be changed to ribosome occupancy.

We have changed the term ‘translation efficiency’ with ‘ribosome occupancy’ throughout.

Reviewer #2:

[…] Overall, the manuscript reports an exciting finding that would be of interest to those interested in gene regulation. However, the current version of the manuscript lacks sufficient validation of the genome-wide datasets on which the entire paper is based. In addition, there are a number of caveats to both the experimental approaches and interpretations that should be addressed. For these reasons, I would only recommend publication in eLife after addressing all of the major concerns listed below.

1) The authors' measure of "translational efficiency" used for all analyses is poorly chosen and can be more accurately described as a measure of "ribosome occupancy". Rather than collecting many fractions from across a sucrose gradient in order to infer the number of ribosomes per mRNA (as done in some published methods, e.g. Wang et al. Mol Sys Biol 2016), the authors only collected the "free", light and heavy polysomes. Even with these three pieces of data, the authors combined the light and heavy polysome data for unknown reasons and calculated "translational efficiency" as simply (light+heavy)/(light+heavy+free). The result is a metric that is extremely insensitive to changes in translation rate, since only if an mRNA shifts >2-fold in/out of the "free" fraction containing 0-1 ribosomes will an mRNA be labeled as down/up-regulated. This is problematic for the authors' conclusion that >70% of transcripts are unchanged at the translational level in response to stress (Figure 1) and implies that there are likely many more changes in translational efficiency than the authors capture with their method. More generally, the use of this translation metric rather than a superior alternative diminishes the impact of the manuscript. The authors should clearly explain somewhere in the manuscript that the use of three pools (rather than many small fractions) is sensitive to large changes in translation but relatively insensitive to smaller changes.

As suggested we have changed the term ‘translation efficiency’ with ‘ribosome occupancy’ throughout. We accept the suggestion and added an explanation in the text that “the use of three pools (rather than many small fractions) is sensitive to large changes in translation but relatively insensitive to smaller changes.”

2) There is an underlying assumption of the authors' analyses that is never explicitly stated: That transcript isoforms found to differ in the location of the 5' cap are identical in all other respects. Alternatively, transcripts generated by different promoters might also be spliced differently or use alternative polyadenylation sites, any of which could contribute to the measured translational efficiencies. By attributing any differences in translation to transcription start site, the authors are implying causation from what is only a correlation. In addition to raising this possibility in the text, the authors should undertake some validation experiments for some of their example genes to show that differences in transcription start site alone are sufficient to impart different polysome sedimentation to the mRNAs.

We have now added two examples that demonstrate that the TSS and not other aspects of the transcript, confers sensitivity to eIF4E availability. We used a GFP reporter either under a CMV promoter that initiates at an A nucleotide, or Rpl18 promoter that initiates at a C nucleotide. Upon eIF4E depletion by siRNA or its inactivation by the 4EGI-1 drug, we found a clear differential sensitivity of the same GFP transcripts (see Figure 4). In addition, we provide validation that the isoforms derived from APs of eIF4A and Pabp are differentially translated following stress (Figure 5 and Figure 6). Moreover, using a reporter gene we show that the differential translation is a consequence of the alternative promoter and not other aspects in the endogenous transcript (Figure 5). As suggested we added the possibility of the potential link to other aspects in the transcript to the Discussion (bottom of the first paragraph).

3) Based on the examples of CapSeq reads depicted (e.g., Figure 3), the number of reads mapping to individual genes seems relatively low (though no scale is provided on the y axis). In addition, because the CapSeq method relies on ligation to the 5' ends of RNAs, sequence differences at the 5' end likely affect how efficiently ends are captured by CapSeq. For these reasons, it's important to undertake some validation of the quantitative changes in transcription start site the authors describe, by using alternative methods such as Northern blotting. In addition, the wording used to describe quantitative changes in start-site usage should be toned down from its current form to be more consistent with what the data conclusively demonstrate.

As stated in our response to point 2 above, we now added validation experiments for differential translation using 5’RACE (without ligation) (see Figure 5 and Figure 6). We also added validation experiments, independent of the CapSeq method, showing the importance of the first nucleotide in translation (Figure 4). In addition, we added y-axis scale to each of the browser displays in the figures.

4) From Figure 3 onward, the paper reads more like a collection of cherry-picked anecdotes rather than a systematic study of a biological phenomenon. While many intriguing examples are shown in Figure 3, Figure 5 and Figure 6, there's no indication as to how widespread or representative these examples are on a transcriptome-wide level. The ending of the paper is particularly weak, as the arbitrary examples of Arhgef2 and Hcfc2 are presented without any obvious connection to the biology being explored in the manuscript. These issues should be addressed through a combination of additional analyses and textual changes.

The aim of the specific examples is to demonstrate that the global analysis is valid to specific genes and has potential biological consequences. A subset of those examples such as eIF4A, PABP and the newly added Rpl18 is also followed by experimental validation (Figure 4, Figure 5 and Figure 6). Most of the un-validated examples are shown as Supplementary files (Figure 6—figure supplement 1 and Figure 6—figure supplement 2).

Reviewer #3:

*[…] Comments:*

The strengths of this study are several fold: (I) generation of an important high quality dataset which is useful for scientists interested in further analyses; (II) important insightful information gathered by the analysis described in the manuscript; (III) experimental validation of several high impact examples. The weakness of the manuscript in its present form is the absence of Supplementary excel file Tables containing the information related to the performed analysis (item II). Such tables, containing list of genes grouped and itemized according to specific features described and discussed in this study should be added.

We now provide an excel file containing several sheets with all the described gene lists ([Supplementary-material SD1-data]).

Figure 1 and corresponding text in the second paragraph of the subsection “Marked increase in differential translation of alternative promoters following stress”: although the experimental steps are well explained, it is not clear how the data quantitatively characterize the translational properties of individual TSS. I guess it will be clearer to the translation field readers if the author will add an explanatory note about the reads length and what one can learn from each read. For example, the note at the beginning of the Results section explaining that ribosomal footprinting is not suitable here, is helpful for understanding. I suggest to add an additional explanatory sentence to clarify that downstream sequences shared by transcripts with different TSS cannot be found as separate reads. The fact that all reads contain the TSS itself is not trivial for people used to think about polysome-generated reads in classical ribosomal profiling experiments. A few words about the CapSeq methodology and which kind of reads to expect will help to understand it. In addition to the explanatory text in the Results section, please also add to the Methods section (end of subsection “Analysis of global translation using CapSeq”) the details of the size selection. A small illustration at the bottom of Figure 1 may enhance clarity too.

We now added explanatory sentences for determination of the 5’ends of transcripts in the polysomal profiling and additional details of the size selection procedure in the Methods.

The explanatory text to Figure 1 (subsection “Marked increase in differential translation of alternative promoters following stress”, fourth paragraph) is not clear. Moreover, numbers mentioned in the text do not correspond to the numbers on the figure itself.

We revised the text for better clarity. Thanks for noting the error with the numbers which was corrected.

Legend to Figure 1 – The text in the last 4 lines is not included in figure.

Thanks for this note. Indeed, the last sentence was moved to Figure 1.

Polysomes association may reflect attenuation at the elongation level. See for example Shenton et al. JBC 2006 (showing translation attenuation at the elongation level in response to oxidative stress). This notion should be mentioned in the text. In addition, please add to the legend to Figure 1): "due to translation regulation at both the initiation and elongation levels".

We clarified that the inhibition of translation in response to energy stress is at the levels of initiation and elongation (see Introduction).

Informative details on the genomic locus at the lower part of Figure 5 (exon1, start codon, intron 1, exon2), as well as to Figure 6 and Figure 6—figure supplement 1, should be added.

As suggested we added chromosomal location and genomic structure for each browser display presentation.

Figure 4: p values should be part of Figure 4 (not in the supplement).

The *eLife* style is to place the supplementary figures adjacent to the main figure. As Figure 4 size in the revised paper has substantially increased we prefer this information to be part of the supplement.

Subsection “The TSS nucleotides influence the cap-binding affinity of eIF4E”, last paragraph: should add after 'RNA oligos' "whose first three nucleotides are CCU, ACU or GCU".

Thanks for the suggestion, the phrase was added.

Discussion, second paragraph: The work of Tcherkezian et al. 2014 is incorrectly interpreted. In this paper LARP1 was identified in a proteomic screen for proteins that associate with the mRNA 5′ cap, but was shown to be associated with PABP, not with eIF4E or eIF4G (whereas it does co-immunoprecipitated with eIF4A).

Thanks, this was corrected.

Subsection “Differential translation of transcript isoforms is part of the stress response”, second paragraph – add a reference to the statement that the truncated eIF4A retains its interaction with eIF4G1.

A reference was added.

Discussion, end of fourth paragraph: IRES is not the only possibility. The uAUG-burdened long 5'UTR of eIF4A might confer either IRES-independent or 5'cap-dependnet translation regulation (see review by Young and Wek, JBC 2016).

Thanks for raising this point, we added it to the Discussion.

Discussion, last paragraph: To be specific, must change from stressful conditions to energy stress. Indeed it may imply that a similar mechanism is operative in response to other stressful conditions, but this should be added as a possibility.

Corrected.